# Effect of experimental, morphological and mechanical factors on the murine spinal cord subjected to transverse contusion: A finite element study

Marion Fournely[1,2], Yvan Petit[2,3,4], Eric Wagnac[2,3,4], Morgane Evin[1,2], Pierre-Jean Arnoux[1,2]*

**1** Laboratoire de Biomécanique Appliquée (LBA), UMR T24, Aix-Marseille Université, IFSTTAR, Marseille, France, **2** International Laboratory on Spine Imaging and Biomechanics (iLab-Spine), Marseille, France, **3** Mechanical Engineering Department, École de technologie supérieure, Montréal, Canada, **4** Research Center, Hôpital du Sacré-Cœur, Montréal, Canada

* pierre-jean.arnoux@ifsttar.fr

**Data Availability Statement:** All relevant data are within the paper and its Supporting Information files.

## Abstract

Finite element models combined with animal experimental models of spinal cord injury provides the opportunity for investigating the effects of the injury mechanism on the neural tissue deformation and the resulting tissue damage. Thus, we developed a finite element model of the mouse cervical spinal cord in order to investigate the effect of morphological, experimental and mechanical factors on the spinal cord mechanical behavior subjected to transverse contusion. The overall mechanical behavior of the model was validated with experimental data of unilateral cervical contusion in mice. The effects of the spinal cord material properties, diameter and curvature, and of the impactor position and inclination on the strain distribution were investigated in 8 spinal cord anatomical regions of interest for 98 configurations of the model. Pareto analysis revealed that the material properties had a significant effect (p<0.01) for all regions of interest of the spinal cord and was the most influential factor for 7 out of 8 regions. This highlighted the need for comprehensive mechanical characterization of the gray and white matter in order to develop effective models capable of predicting tissue deformation during spinal cord injuries.

## Introduction

Spinal cord injuries (SCI) affect between 250 000 and 500 000 persons every year [1]. Presently there is no efficient and safe treatment against the neurological consequences of such injuries while tetraplegia and paraplegia represent 33% of spinal cord injuries in United-States [2]. There is social and economic needs to develop efficient treatment or better prevention strategies. To this end, it is necessary to further understand the injury mechanisms of the spinal cord (SC) and consequently investigate injury biomechanics of such tissues and structures.

Indeed, traumatic SCI are initiated by mechanical loading of the cord tissue. The ensuing tissue deformation may locally lead to functional or structural failure of the tissue components

**Funding:** This work was supported by the Institut Français des sciences et technologies des transports, de l'aménagement et des réseaux(MF), the A*MIDEX project (ANR-11-IDEX-0001-02) (MF). The funders had no role in study design, data collection and analysis, decision to publish, or preparation of the manuscript.

**Competing interests:** The authors have declared that no competing interests exist.

such as cells, blood vessels and axons. This initial and immediate mechanical damage is the primary injury of the SCI. Secondary injury is the result of vascular and biochemical processes taking place during the minutes and weeks following the trauma [3]. Clinical investigation of SCI injury mechanisms has mainly focused on the secondary injury mechanisms because of the difficulties associated with the access to the initial mechanical loading conditions and the immediate tissue damage [4]. However, investigation of the relationships between the initial mechanical loading conditions and the tissue mechanical response up to local damage is becoming a requirement to ensure an exhaustive understanding of injury mechanisms in SCI [5]. Animal models of SCI offer the possibility to investigate such relationships through imaging and histology techniques. It has been shown in animals models that the biomechanical parameters of experimental contusion models, such as the compression rate and the amount of deformation energy applied to the spinal cord, strongly correlate with the extent of immediate tissue damage such as loss of grey matter, axonal disruption or hemorrhage volume [6–10].

In addition, finite element (FE) models of SCI have been developed to replicate experimental models and investigate the mechanical stress and strain distribution within the spinal cord during the injury. These numerical models enable investigating the relationship between the mechanical loading and the mechanical response of the spinal cord. The combination of experimental and numerical analyses could, therefore, have the potential to predict acute tissue damage and to establish operational and structural thresholds for spinal cord tissues [5,11]. In particular, combining FE and experimental models have highlighted the correlation between spinal cord tissue damage and maximal principal strain patterns in the rat [11,12] and the non-human primate [13]. Moreover, correlations have been shown between minimum principal strains and the density of surviving neurons in the gray matter in the rat [14].

To date, there is no biomechanical model representing the mouse spinal cord, whereas the mouse is the second most widely used species in SCI animal models [15]. Yet, the mouse spinal cord and its surrounding structures have distinctive morphological and structural features, including its size and its proportion of white matter (WM) and gray matter (GM) [16] or the absence of subarachnoid space [17]. These morphological features have been shown to strongly affect the biomechanics of SCI in numerical models [18]. Thus, accurate representation of the mouse spinal cord structure is required to properly correlate numerical results (local mechanical stresses and strains) with experimental results (local tissue damage).

In addition to morphological considerations, mechanical stress and strain distribution throughout the spinal cord depend on the material properties of the tissue. It has been established that the spinal cord tissues exhibit hyper-elastic (non-linear) and visco-elastic (load rate dependent) behavior [19]. However, the regional mechanical characteristics, in particular with regard to the differences between WM and GM are controversial. Recent studies on the material properties of brain tissues have shown that in the brain, the WM is about twice stiffer than the GM [20–23]. As for the spinal cord, few studies proposed similar protocols to test WM and GM separately. Three studies showed that the GM was stiffer than the WM by tensile testing [24,25] and by indentation [26]. Ozawa et al. found no significant differences between WM and GM mechanical characteristics using pipette aspiration method [27]. Thus, in biomechanical models of the spinal cord, material properties of WM and GM can largely differ depending on the original source of data, leading to potential bias in the simulated local strain and stress throughout the spinal cord.

The aims of this study were twofold. The first objective was to develop a biomechanical model of spinal cord injury in mouse and to validate it against experimental data. The second one was to investigate the influence of morphological and experimental factors as well as regional material properties on the local mechanical responses of the spinal cord in order to improve FE model for the investigation of SCI biomechanics.

## Materials & methods

A detailed FE model of the mouse cervical spinal cord was developed to reproduce an experimental model of dynamic lateral cervical contusion [28]. The model was developed using the HyperWorks suite (Altair Engineering, USA) and solved with the non-linear explicit Radioss solver, release 12.0 (Altair Engineering, USA)

### Model generation

The model geometry was built from the Allen Spinal Cord Atlas [29] and MRI images of mouse cervical spinal cord [30]. Such atlas images were used to delineate transverse cross-sections of the spinal cord from C1 to C8 and to distinguish white matter (WM) from gray matter (GM) (Fig 1A). WM and GM were each delineated in four sub-regions of interest (ROIs). On each cross-section 8 ROIs were delineated: dorsal WM, ventral WM, ipsi-lateral (left side) WM, contra-lateral (right side) WM, dorsal ipsi-lateral GM, ventral ipsi-lateral GM, dorsal contra-lateral GM and ventral contra-lateral GM (Fig 1B). The cross sections were uniformly scaled so that C4 anteroposterior diameter was 1.45 mm as observed in the replicated experimental model [28]. Sagittal MR acquisitions [30] were used to orient and place the cross-section. However, these images were acquired with a vertical imager (Brucker, Ettlingen, Germany), resulting in an extended posture of the mice cervical spine (Fig 1C). Two additional postural configurations of the model were also generated: one straight and one reversing the MRI-based curvature (Fig 1D). Pia and dura maters were modeled as thin shells bonded to the spinal cord with a 0.01 mm thickness as observed in histological studies of mouse meninges [17]. The vertebral foramen was defined by its inner surface that was built as a scaling of the meninges (Fig 1E). The impactor tip was modelled as a cylindrical surface of 0.6mm diameter with rounded edges and was positioned at the level of the dorsal left horn of the gray matter at C4 vertebral level (Fig 1E).

GM and WM were meshed with linear tetrahedral elements having one integration point (141k elements, with an element size of 0.1 mm). Meninges and vertebral foramen were meshed with fully-integrated 3-nodes shell elements (19k elements, with an element size of 0.08 mm). The impactor was meshed with fully-integrated 4-nodes shell elements (2k elements with an element size of 0.03 mm).

The meninges properties were extracted from an experimental study on the material properties of the rat dura mater [31]. WM and GM hyper-elastic mechanical behaviors were modeled using $1^{st}$ order Ogden laws and their viscoelastic behavior was modeled by Prony series. Hence, the strain energy density (W) was computed with the following equation:

$$W = \frac{\mu}{\alpha} \left( \bar{\lambda}_1^\alpha + \bar{\lambda}_2^\alpha + \bar{\lambda}_3^\alpha - 3 \right) + \frac{K}{2} \left( J - 1 \right)^2$$

Where $\mu$ and $\alpha$ are material constants and with $\bar{\lambda}_i = J^{1/3} \lambda_i$, $\lambda_i$ being the $i^{th}$ principal stretch and $J = \lambda_1 \times \lambda_2 \times \lambda_3$ being the relative volume, with $K = \mu \times \alpha (1+v)/3 (1-2v)$ being the bulk modulus and $v$ being the Poisson's ratio.

Viscoelastic properties were introduced considering Prony series:

$$G(t) = \frac{\alpha \times \mu}{2} + \sum_{k=1}^{n} G_k \, exp^{-t/\tau_k}$$

Where $G$ is the shear modulus, $\mu$ and $\alpha$ the Ogden material constants, $\tau_k$ are the relaxation times and $G_k$ the shear moduli associated with them.

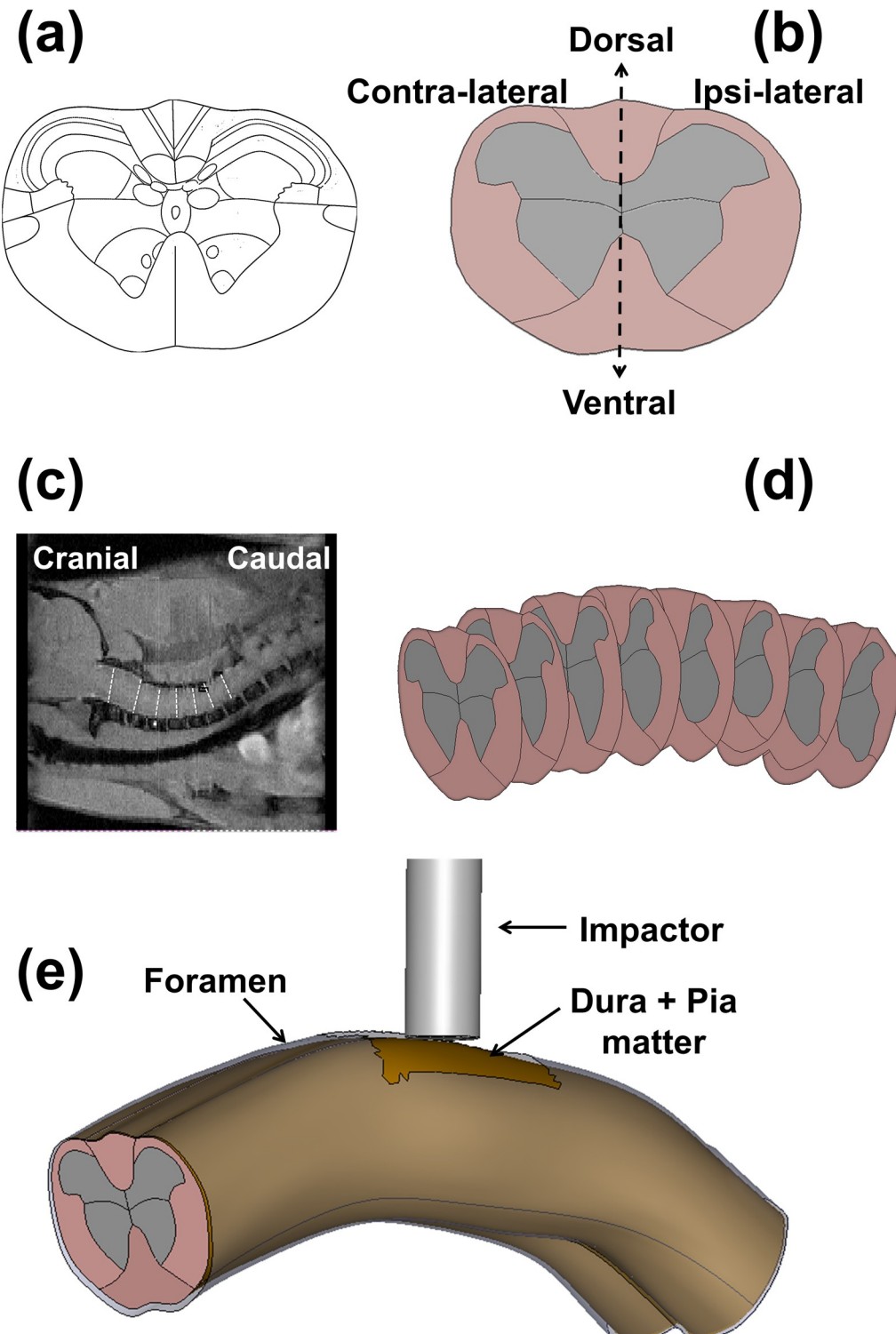

**Fig 1. Mouse spinal cord finite element model geometry.** (a) Structural map of the spinal cord at the C1 vertebral level (adapted from the Mouse Spinal Cord Atlas [29]); (b) Reconstruction of the slice with delimitation of 8 regions of interest: dorsal, ventral, ipsilateral and contralateral white matter and ipsi-dorsal, ipsi-ventral, contra-dorsal and contra-ventral gray matter; (c) MRI acquisition [30] of the spinal cord in the sagittal plane allowing position and orientation assessment for each slice at each vertebral level; (d) Inverse MRI-based curvature model of the cervical spinal cord from C1 to C8; (e) 3D model of the SC with meninges, vertebral foramen canal and impactor.

**Table 1. Summary of the material properties used in the different models (WM = white matter and GM = gray matter).** Ogden and Prony parameters are defined here according to Radioss notation for compatibility with the solver, and may differ from the notation used in the referenced papers.

| | | Poisson's ratio | Ogden parameters | | Bulk Modulus | Prony Serie parameters | | Reference |
|---|---|---|---|---|---|---|---|---|
| | | | $\mu$ (MPa) | $\alpha$ | (MPa) | $G_i$ (MPa) // $\tau_i$ (ms) | | |
| **For all models** | Dura | 0.45 | 1.48 E-01 | 16.2 | 1.16E+01 | $G_1 = 1.069$ | $\tau_1 = 9$ | [31] |
| | | | | | | $G_2 = 0.416$ | $\tau_2 = 81$ | |
| | | | | | | $G_3 = 0.335$ | $\tau_3 = 564$ | |
| **Model WM > GM** | WM | 0.45 | 3.3 E-02 | 3.99 | 6.36E-01 | $G_1 = 0.209$ | $\tau_1 = 1E+02$ | [13] |
| | | | | | | $G_2 = 0.113$ | $\tau_2 = 1E+03$ | |
| | | | | | | $G_3 = 0.061$ | $\tau_3 = 1E+04$ | |
| | | | | | | $G_4 = 0.033$ | $\tau_4 = 1E+05$ | |
| | GM | 0.45 | 1.46 E-03 | 7.52 | 5.31E-02 | $G_1 = 1.069$ | $\tau_1 = 6.4E+02$ | |
| | | | | | | $G_2 = 0.416$ | $\tau_2 = 6.4E+03$ | |
| | | | | | | $G_3 = 0.335$ | $\tau_3 = 6.4E+04$ | |
| **Model GM > WM** | WM | 0.45 | 4.7 E-04 | 17.36 | 3.94E-02 | // | // | [24] |
| | GM | 0.45 | 7.7 E-04 | 19.6 | 7.29E-02 | // | // | |
| **Model GM = WM** | WM & GM | 0.45 | 1.4 E-02 | 4.7 | 3.18E-01 | $G_1 = 0.099$ | $\tau_1 = 8$ | [11] |
| | | | | | | $G_2 = 0.056$ | $\tau_2 = 150$ | |

Three different data sets were used to model WM and GM properties as observed in the literature. First, 1st order Ogden laws were fitted to Ichihara et al. [22] experimental data. These data were derived from tensile tests on bovine WM and GM distinctly and present a gray matter twice stiffer than white matter. Secondly, the laws developed by Maikos et al. [11] for describing the rat spinal cord mechanical behavior were used. In this study, the spinal cord was described as a uniform structure and white and gray matters had the same stiffness. Thirdly, according to a recent work characterizing the white matter on non-human primates [13], the white matter material properties were considered stiffer than the gray matter. The different sets of material properties are summarized in Table 1. Their corresponding bulk modulus (ranging from 3.94E-1 MPa to 6.36E-1 MPa) is indicated as a representative measure of how resistant to compression each material is.

The impactor and the vertebral foramen were defined as rigid bodies. The extremities of the meninges and the spinal cord were constrained in rotation in the transverse plane and in translation and in the cranio-caudal direction. A dorsoventral displacement was applied to the impactor at a speed of 0.12 mm/ms and a peak displacement of 0.725 mm corresponding to a compression ratio of 50%.

## Validation of the model

The process of validation of the FEM was supported by previously published experimental data [28]. Female 17–30 weeks old C57Bl/6 mice (n = 9) underwent a partial dorsolateral laminectomy of the left C4 lamina. The spine was immobilized with two clamps positioned on C3 and C5 vertebra. Mice underwent a 30 kDyn (0.3N) lateral cervical contusion with the Infinite Horizon Impactor (Precision System and Instrument LLC, US) equipped with a 0.6mm diameter tip. The tip vs. time and the rectified applied compression vs. time [28] were recorded at 1 kHz to extract between 17 and 29 data points for each acquisition. The nine experiments enabled a definition of an experimental corridor of force-displacement curves. The experimental data points and corridor are presented in Fig 2. The spinal cord force-displacement response was compared between experimental data and FE simulations with the three sets of material properties.

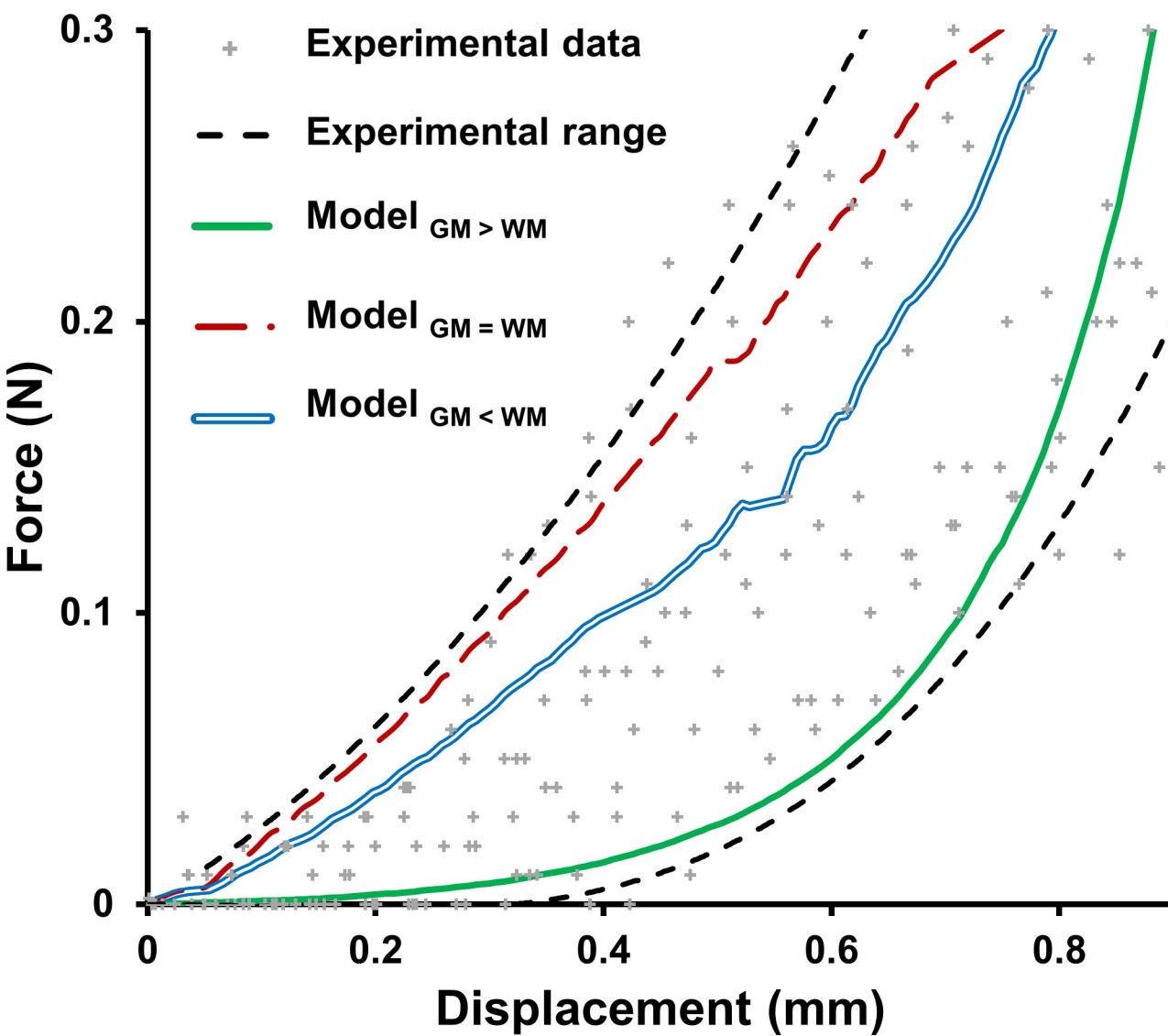

**Fig 2. Model validation.** Comparison of the experimental results [28] of force–displacement relationship for a lateral cervical contusion with a 50% compression ratio and the corresponding numerical results of FE model using three data sets derived from available literature for white matter and gray matter mechanical properties.

### Analysis of the effects of morphological, experimental and mechanical factors

The FE model was configured to test the effects of seven different factors. For each factor, three levels were defined: one neutral and two to represent the possible variation of that factor.

Two morphological factors were tested: the curvature and the diameter of the spinal cord. As previously described, three different curvatures were defined: an extended posture based on MRI acquisition, a straight posture and a flexed posture (obtained by reversing the extended posture) (Fig 1D). Three cord diameters were tested (1.45 ± 0.16 mm) to account for intraspecies variability [28].

Four experimental factors were tested: the impactor cranio-caudal and lateral positions and the impactor sagittal and transverse tilt. Accordingly, the impactor was translated laterally and

cranio-caudally of ±0.1mm from its initial position and rotated in the transverse plane (± 3˚) and in the sagittal plane (± 3˚).

The different data sets for WM and GM material properties described in *Model Generation* were tested as a three-level mechanical factor: one with WM stiffer than GM, one with GM stiffer than WM and one with homogeneous WM and GM.

To summarize, the effects of 7 three-level factors were investigated: material properties, SC curvature, SC diameter, impactor transverse tilt, impactor sagittal tilt, impactor lateral positioning and impactor cranio-caudal positioning. Each factor was individually tested and 11 interactions of interest were also tested (listed in Table 2) This investigation resulted in a total of 98 runs.

An analysis of variance was performed using Statistica 13 (StatSoft, Inc., USA) to study the main effects of the factors mentioned above and some of their interactions for each of the following outcome measurements: mean maximal principal strain, shear strain and VM strain of each ROI. Mean strains were obtained by averaging strains on a set of elements at the impacted site. The set was defined by all the elements included in a 0.6 mm cross-sectional slice centered at C4 vertebral level (ie. under the impactor) resulting in about 13k elements.

Due to the determinist aspect of FEM simulations, an alpha acceptance of less than 0.01 was chosen for significance.

## Results

The computed force-displacement curves of the three FE model configurations with the different material data sets all laid within the range of experimental results and showed the distinctive J-shaped curvature of experimental data. (Fig 2). This shows that the three material data sets allowed for an accurate representation of the overall mechanical behavior of the mouse spinal cord under dynamic compression up to a 50% compression ratio.

The strain distribution throughout the spinal cord was strongly affected by the relative change in material properties between WM and GM in the model (Fig 3). The numerical results presented in this paragraph compare the peak values of the maximal principal strain in each ROI for models with neutral positioning of the impactor, mean cord diameter and straight posture of the spine and variable sets of material properties. Peak values were determined as the 95[th] percentile of maximal principal strain of all elements within a ROI.

Representing the GM stiffer than WM led to a more diffused strain field throughout the spinal cord (as shown in the sagittal sections of Fig 3) and to higher strain levels in the ventral part of the WM (85% vs. 49% and 17%). The uniform properties between WM and GM led to a strain field focused on the impact site, and low strain levels in the contra-lateral and ventral part (49% vs. 85% and 65%). Representing stiffer WM than GM led to higher strain levels in the contra-lateral GM (81% vs 41% and 43%) and ipsi-lateral GM (120% vs. 113% and 80%).

**Table 2. Interactions studied between the different factors.**

|  | SC Diameter | Sagittal Tilt | Transverse Tilt | Spine Curvature | Cr-Cau Position | Lateral Position | Material Prop. |
|---|---|---|---|---|---|---|---|
| **SC Diameter** |  |  |  |  |  | x | x |
| **Sagittal Tilt** |  |  |  |  |  | x | x |
| **Transverse Tilt** |  |  |  |  |  | x | x |
| **Spine Curvature** |  |  |  |  | x | x | x |
| **Cr-Cau Position** |  |  |  | x |  |  | x |
| **Lateral Position** | x | x | x | x |  |  | x |
| **Material Prop.** | x | x | x | x | x | x |  |

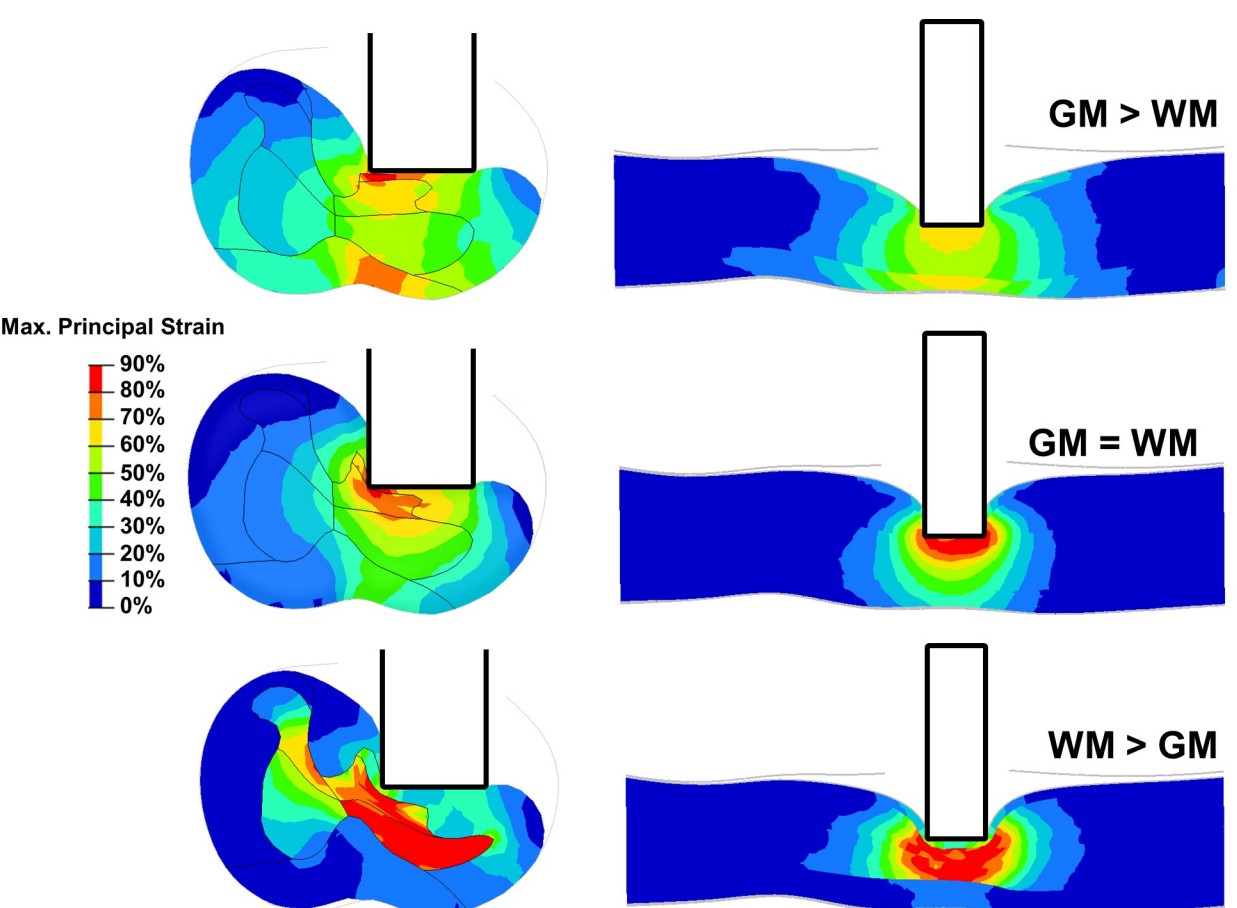

**Fig 3. Comparison of strain distribution.** Maximal principal strains throughout a cross-section (left) and a sagittal section (right) of the spinal cord at mid impactor for the three different material data sets.

This set of material properties also led to lower levels of strain in the ipsilateral WM (56% vs. 96% and 90%).

The analysis of variance allowed identifying the factors with significant effects on the maximal principal strain, maximal shear strain and VM strain, for each ROI (Fig 4). However, only the main effects on the maximal principal strain and their interactions are presented for each ROI in Fig 4 since the same factors have a significant effect on the shear strain and the VM strain (S1 and S2 Figs). It reveals that the material properties have a significant effect (p<0.01) for all ROI and this is the most influential factor for 7 out of 8 eight ROIs. Lateral positioning of the impactor has a significant effect on 7 ROIs and SC diameter on 5 ROIs. Transverse tilt significantly affected 2 ROIs and the other principal factors show no significant effect for any ROI.

## Discussion

To our knowledge, this work presents the first murine FE model of spinal cord contusion. The numerical model was validated against experimental data from a corresponding mouse spinal cord injury experimental model. The developed model demonstrated its ability to simulate the mechanical behavior of the mouse spinal cord under transverse contusion. The influence of mechanical, experimental and morphological factors on the numerical model was assessed by

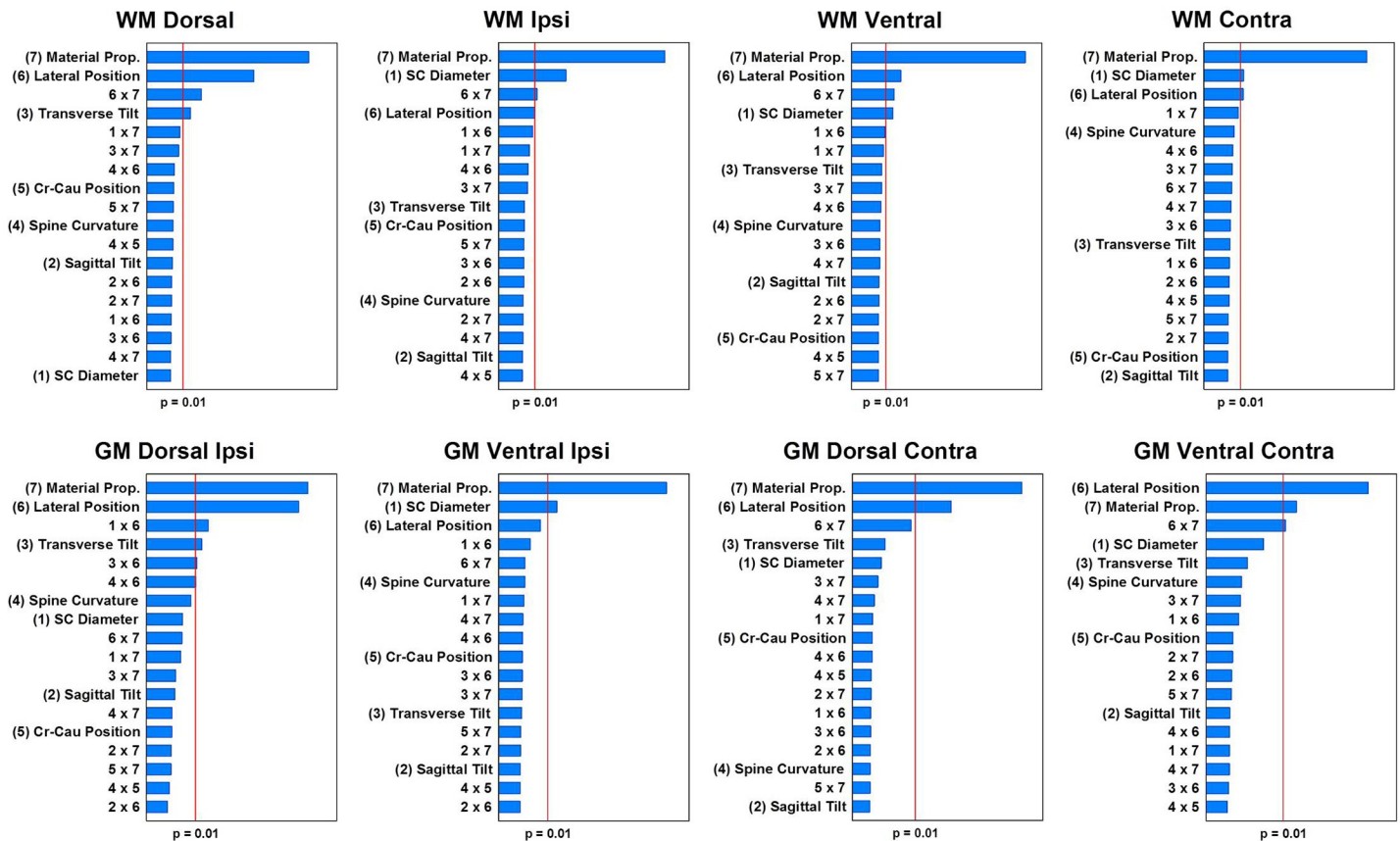

**Fig 4. Pareto charts of the standardized effects on maximal principal strain for each region of interest.** Coupled effects are noted "n × m", with "n" and "m" the factor denomination. The standardized effects of each factor are presented in descending order of influence on the maximal principal strain.

analyzing the computed tissue deformation throughout the cord. The identification of such key factors allows for the definition of action priorities in the investigation of SCI biomechanics. This work highlights that the material properties assigned to the white and gray matter can strongly affect the simulated local strains throughout the spinal cord during a trauma. Thus, the first priority in future works should be a comprehensive characterization of the material properties of WM and GM in order to develop models predicting tissue deformation during spinal cord injuries.

The FE model was validated against previous experimental data. For the three sets of material properties, its overall mechanical behavior laid within our experimental range. This shows that regardless of the assumptions made on the WM and GM material properties, it is possible to simulate the global mechanical behavior of the spinal cord. The three sets were chosen in view of the available data and have all been used in spinal cord FE model investigating injury mechanisms [12,13,18,32–34]. This study demonstrated the strong impact of the choice of material properties on the local mechanical behavior throughout the spinal cord. The use of a FE model to predict strain patterns–and thus tissue damage patterns—should require stronger validation than the overall mechanical behavior of the model. In the absence of a consensus on the material properties of white and gray matter in the spinal cord, providing sensitivity analysis on these properties should address this lack of knowledge.

The major impact of the WM and GM material properties on the strain distribution was acknowledgeable both statistically and qualitatively. This result is consistent with previous

biomechanical studies which have highlighted the sensitivity of SCI FE models to their material properties variations [11,35]. The method presented in this study stood out by a comprehensive definition of regions of interest, allowing for specific analysis on each sub-region of the WM and GM. Notably, for a unilateral injury, such as the one introduced, an important issue is to spare the contralateral part of the cord [36]. Changes in material properties affected the level of strain in both the contralateral WM and GM. In particular, inhomogeneous WM and GM properties led to higher levels of strain in the contralateral part of the cord. Experimental models of unilateral contusion at the mid-cervical level in mice have demonstrated minimal tissue damage in the contralateral part of the cord, but also motor function deficits in the contralateral limbs [37,38]. These experimental observations together with an accurate simulation for computation of the strain fields within the spinal cord could allow for the definition of mechanical thresholds for spinal cord tissues. This underlines the need for additional experimental investigation of the WM and GM material properties.

Beyond the major effect of material properties, experimental factors have been shown to significantly affect the strain levels within the spinal cord. Notably, lateral positioning of the impactor had a significant effect on the strain levels in the ventral parts of the WM and the GM. Considering that higher levels of strain within a SC region may cause greater acute tissue damage, the exact lateral position of the impactor could lead to greater damage in the ventral WM and the ventral GM. In the rodents, and at the cervical level, the locomotor-related motor neurons and descending tracts are concentrated in these particular ROIs [39]. In experimental models, lateral positioning of the impactor could lead to different levels of motor function impairment. Therefore, specific attention should be given to this factor in experimental set-ups in order to ensure their reproducibility.

The other experimental factors (tilt of the impactor and its position relative to the cranio-caudal axis of the spinal cord) investigated in this study did not appear to significantly affect the local mechanical behavior of the spinal cord. This maybe explained in part by the strain values that were averaged in each ROI, which could have neglected strain concentrations and lessen the effect of the impactor positioning. In experimental models, strain concentrations could correspond to structural or physiological damage at localized areas of the cord. As a consequence, the reproducibility of the impactor positioning should be ensured as far as practicable in experimental studies. As for morphological variables, the effect of the spinal cord curvature was negligible, which suggests there is no need to assess the curvature of the spinal cord during experimental trauma. The significant effect on the spinal cord diameter on the strain levels was an expected result as for different spinal cord sizes, a similar loading condition will lead to different compression levels. The constant displacement imposed to the impactor led to a compression ratio of 45% and 56% for the cord with the largest diameter and the small diameter respectively.

This study design involved several assumptions. First, the sets of material properties describing WM and GM mechanical behavior were selected to be representative of the actual state of the art in SCI FE analysis. This set of material properties did not cover the entire range of possible WM and GM mechanical behaviors and did not take into account the anisotropic behavior of the white matter [40]. However, the important effect of the material properties found in this study highlights the need for comprehensive and distinct characterization of the material properties of WM and GM. Secondly, the effects of several factors were not investigated such as the boundary conditions, the length of the modeled cord or contact formulation. These choices were motivated by the fact that these factors have already been assessed in a FE model of the rat spinal cord and were found not to cause significant differences in strain distribution [11]. Further study could include the interactions between the spine curvature, the cord diameter, and the impactor tilt which were not investigated in this analysis. Thirdly, the

present FE model aimed to reproduce a specific experimental model: a unilateral contusion at the C4 cervical level with an impactor tip of 0.6 mm diameter reaching a 50% compression ratio of the spinal cord. It has been validated in this study for transverse compression loading with compression ratios ranging between 0 and 50% and at high strain rates (up to 80 s$^{-1}$). Thus, the use of the model for other loading conditions in term of injury magnitude should be accompanied by further validation. The model maybe used to simulate contusion at different cervical levels, with special care to the effect of boundary conditions. The factors affecting the different variables may differ with different loading conditions. However, the strong difference between the effect of material properties and the other factors suggests a significant effect of the material properties on the strain distribution regardless of different loading conditions. Lastly, we chose to assess only strains rather than stress as the latter are related to the material stiffness. As the different material properties used for modeling WM and GM displayed strong differences in stiffness, analyzing the effects on stresses would have led to an exacerbation of the effect of material properties.

In conclusion, the investigation of the SCI primary damage through FE model shows promising potential, but there is a tremendous need for additional characterization of the mechanical behavior of white and gray matters. While such additional characterization should enable the investigation of the relationships between biomechanical parameters, local mechanical behavior and tissue damage in SCI, the results presented in this work enable to quantitatively describe the influence of changes in material properties. The model and its associated modeling choices presented in this study allowed computing of the overall mechanical response of the spinal cord under traumatic injury and thorough assessment of the effect of experimental and morphological factors on the local mechanical behavior throughout the cord. Sensitive experimental factors such as the positioning of the impactor were identified. Under the assumption that slight changes in local mechanical strains may initiate tissue damage, the uncertainty on these factors in experimental models could account for a lack of reproducibility. This highlights the fact that FE models could provide recommendations to help improve the repeatability and reproducibility of experimental models.

## Supporting information

**S1 Fig. Von mises strain mean values for each level of each principal factor.** Values were obtained from WM ROIs(A) and GM ROIs(B). p-value was assessed through Wilcoxon test. $^{***}$: p<0.001, $^{**}$:p<0.01, $^{*}$:p<0.5.
(TIF)

**S2 Fig. Shear strain mean values for each level of each principal factor.** Values were obtained from WM ROIs (A) and GM ROIs (B). p-value was assessed through Wilcoxon test. $^{***}$: p<0.001, $^{**}$:p<0.01, $^{*}$:p<0.5.
(TIF)

## Author Contributions

**Conceptualization:** Marion Fournely, Yvan Petit, Eric Wagnac, Pierre-Jean Arnoux.

**Data curation:** Marion Fournely, Morgane Evin.

**Formal analysis:** Marion Fournely, Morgane Evin.

**Funding acquisition:** Pierre-Jean Arnoux.

**Investigation:** Marion Fournely, Yvan Petit, Eric Wagnac, Morgane Evin, Pierre-Jean Arnoux.

**Methodology:** Marion Fournely, Yvan Petit, Eric Wagnac, Morgane Evin, Pierre-Jean Arnoux.

**Project administration:** Yvan Petit, Eric Wagnac, Pierre-Jean Arnoux.

**Resources:** Yvan Petit, Eric Wagnac, Pierre-Jean Arnoux.

**Software:** Marion Fournely.

**Supervision:** Yvan Petit, Eric Wagnac, Pierre-Jean Arnoux.

**Validation:** Marion Fournely, Yvan Petit, Eric Wagnac, Morgane Evin, Pierre-Jean Arnoux.

**Visualization:** Marion Fournely, Morgane Evin.

**Writing – original draft:** Marion Fournely.

**Writing – review & editing:** Marion Fournely, Yvan Petit, Eric Wagnac, Morgane Evin, Pierre-Jean Arnoux.

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
