## [Decision Letter · Decision Letter 0]

10 Mar 2020

PONE-D-20-03400

Effect of experimental, morphological and mechanical factors on the murine spinal cord subjected to transverse contusion : a finite element study

PLOS ONE

Dear Mrs. Fournely,

Thank you for submitting your manuscript to PLOS ONE. After careful consideration, we feel that it has merit but does not fully meet PLOS ONE’s publication criteria as it currently stands. Therefore, we invite you to submit a revised version of the manuscript that addresses the points raised during the review process.

Notably, since previous work is utilized, more details are requested on experimental data presented with pertinent citations. The reviewers ask you to discuss certain aspects of your findings in relation to reproducibility and/or applicability of your chosen model.  This is a rather technical report and an overview of the highlights is recommended for the broader readership, whether in abstract or introduction/discussion. Notably, the sentence "factors on the spinal cord behavior subjected to transverse contusion" seems inappropriate when no behavior of the animals is assessed in this report. 

We would appreciate receiving your revised manuscript by Apr 24 2020 11:59PM. To enhance the reproducibility of your results, we recommend that if applicable you deposit your laboratory protocols in protocols.io, where a protocol can be assigned its own identifier (DOI) such that it can be cited independently in the future. For instructions see: http://journals.plos.org/plosone/s/submission-guidelines#loc-laboratory-protocols

We look forward to receiving your revised manuscript.

Kind regards,

Alexander Rabchevsky, Ph.D.

Academic Editor

PLOS ONE

Journal Requirements:

Reviewers' comments:

Reviewer's Responses to Questions

**Comments to the Author**

1. Is the manuscript technically sound, and do the data support the conclusions?

Reviewer #1: Yes

Reviewer #2: Yes

2. Has the statistical analysis been performed appropriately and rigorously? 

Reviewer #1: Yes

Reviewer #2: Yes

3. Have the authors made all data underlying the findings in their manuscript fully available?

Reviewer #1: Yes

Reviewer #2: Yes

4. Is the manuscript presented in an intelligible fashion and written in standard English?

Reviewer #1: Yes

Reviewer #2: Yes

5. Review Comments to the Author

Reviewer #1: In this manuscript, the authors attempt to develop a biomechanical model of SCI in the mouse, as well as investigate the influence of morphological and experimental factors on induced SCI. This is a worthwhile study and the amount of work t undertake these studies are highly appreciated. Towards these goals the authors validate the devised model against previous work utilizing actual mice SCI performed by the group. Indeed, the authors find that there is promise towards developing an effective model but more work needs to be performed.

Minor concerns are noted:

While the authors do utilize previous work, it would be helpful to provide more details (sex, age, actual experimental results, etc.) on these prior experiments so as not to require the reader to search for these studies.

The author should provide some further discussion on potential outcomes if injury magnitude and location were to change. How would this impact the model? Is the generated model only good for lateralized cervical contusion injuries?

Reviewer #2: The current manuscript examines finite element models of mouse spinal cord injury. The model applies specific strain properties to 8 different areas of the spinal cord and tests different properties for gray vs. white matter areas of the cord. The validity of the models was then tested against force x displacement readouts for previously published unilateral mouse spinal cord injury. The data is novel and predicted values fall within actual experimental results. It is interesting to note that the model falls within experimental results regardless of whether the assumptions of gray vs. white matter strain properties are reversed (gray> white and gray <white) equal="" or="">>>> than white, does the model still fall within experimental results. Regardless the work is likely of interest to investigators examining the biomechanics of spinal cord injury.

  </white)>

6. PLOS authors have the option to publish the peer review history of their article (what does this mean?). If published, this will include your full peer review and any attached files.

Reviewer #1: No

Reviewer #2: No

---

## [Author Response · Author response to Decision Letter 0]

20 Apr 2020

Dear Academic Editor and Reviewers, 

Thank you for the consideration of our manuscript. We appreciate your inputs which will help the manuscript to reach a wider readership. We would like to thank the reviewers for their careful review of the manuscript. Each point was carefully taken into consideration and answered in this letter and the revised version of the manuscript. Changes in the manuscript are highlighted in red. 

Response to the Academic Editor : 

• This is a rather technical report and an overview of the highlights is recommended for the broader readership, whether in abstract or introduction/discussion.

Lines 16-19 and 27-30 of the abstract were revised so that both the methodologies and the conclusions are more straightforward and accessible for a broader readership. 

Opening of the discussion (lines 247-257) was also revised to better highlight the results and their meanings.

• Notably, the sentence "factors on the spinal cord behavior subjected to transverse contusion" seems inappropriate when no behavior of the animals is assessed in this report.

Line 20 and line 144: The term « behavior » was replaced with «mechanical behavior » " to ensure that there can be no ambiguity on the field of study.

Response to Reviewer #1

• While the authors do utilize previous work, it would be helpful to provide more details (sex, age, actual experimental results, etc.) on these prior experiments so as not to require the reader to search for these studies.

Lines 163-165: More information on the experimental model was provided in the manuscript: sex and age of the animals, as well as details regarding the surgical procedure and the experimental setup.

Lines 170: A sentence was added to inform the reader that all experimental data are provided in Fig2.

• The author should provide some further discussion on potential outcomes if injury magnitude and location were to change. How would this impact the model? Is the generated model only good for lateralized cervical contusion injuries?

Line 322-326 : Potential outcomes were provided in the “Limitations” part of the Discussion. The model was validated within a range of loading conditions and therefore could be used under different loading conditions as long as they fall within this range. In particular, simulation of non-lateralized contusion could be carried out as this loading mode is the same as the one validated. However, simulation of a contusion with a compression higher than 50% would require further validation. 

Response to Reviewer #2

• It is interesting to note that the model falls within experimental results regardless of whether the assumptions of gray vs. white matter strain properties are reversed (gray> white and gray >>> than white, does the model still fall within experimental results.

Indeed, the three chosen sets of material properties were chosen to cover plausible range and ratios of WM and GM mechanical behaviors reported in the literature. Exacerbating the ratio between gray and white matters (for example increasing significantly gray matter stiffness as compared to white matter as suggested in your comment) may obviously increase the global stiffness of the cord and potentially fall outside of the experimental corridor. What we know is that it will strongly affect the strain and stress distribution within the cord. This actually, as stated in the discussion (line 263-266), would further demonstrate the strong impact of the choice of material properties and the need for further experimental investigations to better described the WM and GM properties. A sentence was added in the Discussion (lines 259-261), highlighting your observation that it is possible to simulate the global mechanical behavior of the spinal cord regardless of the gray vs white matter ratios tested in this study.

---

## [Editor Report · Decision Letter 1]

27 Apr 2020

Effect of experimental, morphological and mechanical factors on the murine spinal cord subjected to transverse contusion : a finite element study

PONE-D-20-03400R1

Dear Dr. Fournely,

We are pleased to inform you that your manuscript has been judged scientifically suitable for publication and will be formally accepted for publication once it complies with all outstanding technical requirements.

With kind regards,

Alexander Rabchevsky, Ph.D.

Academic Editor

PLOS ONE
---

## [Editor Report · Acceptance letter]

30 Apr 2020

PONE-D-20-03400R1 

Effect of experimental, morphological and mechanical factors on the murine spinal cord subjected to transverse contusion: a finite element study 

Dear Dr. Fournely:

I am pleased to inform you that your manuscript has been deemed suitable for publication in PLOS ONE. Congratulations! Your manuscript is now with our production department. 

With kind regards,

on behalf of

Dr. Alexander Rabchevsky 

Academic Editor

PLOS ONE